# The Impact of Post-Mastectomy Radiotherapy on Survival Outcomes in Breast Cancer Patients Who Underwent Neoadjuvant Chemotherapy

**DOI:** 10.3390/cancers13246205

**Published:** 2021-12-09

**Authors:** Janghee Lee, Jee-Ye Kim, Soong-June Bae, Yeona Cho, Jung-Hwan Ji, Dooreh Kim, Sung-Gwe Ahn, Hyung-Seok Park, Seho Park, Seung-Il Kim, Byeong-Woo Park, Joon Jeong

**Affiliations:** 1Department of Surgery, Gangnam Severance Hospital, Yonsei University College of Medicine, Seoul 06273, Korea; Doctorlee85@outlook.kr (J.L.); mission815815@yuhs.ac (S.-J.B.); SHEVCHENCKO@yuhs.ac (J.-H.J.); rlaenfpd@gmail.com (D.K.); asg2004@yuhs.ac (S.-G.A.);; 2Department of Surgery, Sacred Heart Hospital, Hallym University, Dongtan 18450, Korea; 3Department of Surgery, Severance Hospital, Yonsei University College of Medicine, Seoul 03722, Korea; jeeye0531@yuhs.ac (J.-Y.K.); imgenius@yuhs.ac (H.-S.P.); psh1025@yuhs.ac (S.P.); skim@yuhs.ac (S.-I.K.); bwpark@yuhs.ac (B.-W.P.); 4Institute for Breast Cancer Precision Medicine, Yonsei University College of Medicine, Seoul 06273, Korea; 5Department of Radiation Oncology, Gangnam Severance Hospital, Yonsei University College of Medicine, Seoul 06273, Korea; IAMYONA@yuhs.ac

**Keywords:** breast cancer, neoadjuvant chemotherapy, post-mastectomy radiotherapy, neoadjuvant response index, propensity score matching

## Abstract

**Simple Summary:**

The benefits of post-mastectomy radiotherapy (PMRT) may vary depending on patients’ responses to neoadjuvant chemotherapy (NAC), although PMRT is useful for patients who underwent NAC. One can consider omitting PMRT in patients who have achieved pathologic complete response or who have minimal residual disease, with a neoadjuvant response index value of 0.7–1.0.

**Abstract:**

This study aimed to determine whether post-mastectomy radiotherapy (PMRT) is beneficial for the prognosis of patients who achieved pathologic complete response (pCR), or who had minimal residual disease, after undergoing neoadjuvant chemotherapy (NAC). Patients who underwent a total mastectomy between 2006 and 2018, after NAC, were included. Patients who did not receive PMRT were matched using 1:3 propensity score matching (PSM). Kaplan–Meier survival curves were used to compare locoregional recurrence-free survival (LRRFS) and overall survival (OS). A total of 368 patients were included after 1:3 PSM. PMRT improved the LRRFS (*p* = 0.016) and OS (*p* = 0.017) rates of patients who underwent NAC. However, PMRT did not affect the prognosis of patients with pCR (LRRFS: *p* = 0.999; OS: *p* = 0.453). In addition, PMRT had a limited effect on LRRFS and OS in patients who responded well to NAC, with a neoadjuvant response index (NRI) value of 0.7–1.0 (LRRFS: *p* = 0.568; OS: *p* = 0.875). PMRT improved the OS of patients with a large residual tumor burden, such as nodal metastases or pathologic stage II/III. The benefits of PMRT vary depending on the patients’ response to NAC, although PMRT is useful for treating patients who underwent NAC. PMRT can be omitted, not only in patients with pCR, but also in good responders with an NRI value of 0.7–1.0.

## 1. Introduction

Neoadjuvant chemotherapy (NAC) is widely performed in patients with breast cancer [1,2]. NAC is not different from adjuvant chemotherapy in terms of prognosis, and the response to chemotherapy can be used as a basis for predicting the patient’s prognosis [3,4,5]. Furthermore, additional adjuvant treatment can be considered in patients with residual disease, particularly in those with human epidermal growth factor receptor 2 (HER2)-positive or triple-negative breast cancer [6,7,8,9].

Post-mastectomy radiotherapy (PMRT) reduces locoregional recurrence (LRR) and improves survival outcomes in high-risk patients. In the meta-analysis performed by the Early Breast Cancer Trialist Collaborative Group, PMRT reduced the LRR and breast cancer-specific mortality in patients with one to three metastatic lymph nodes (LNs), as well as those with four or more metastatic LNs [10,11]. The Danish Breast Cancer Cooperative group found that irradiation of the chest wall and regional LNs significantly improved the disease-free survival and overall survival (OS) rates of patients who had undergone mastectomy for stage II or III breast cancer [12]. Existing guidelines recommend PMRT for patients with a tumor diameter of >5 cm (pT3–4), or more than four metastatic LNs (pN2–3). Recently, PMRT has been considered in patients with one to three metastatic LNs (pN1) [13,14,15,16,17].

However, although no definite criteria have been established regarding the need for PMRT after NAC, several retrospective studies have performed this treatment [18,19,20,21]. As patients with a good response to chemotherapy have a better prognosis, the need for irradiation in these patients should be discussed, although existing guidelines recommend that the need for PMRT should be based on the pre-NAC staging [22]. Therefore, we aimed to determine whether PMRT is beneficial for the prognosis of patients who achieved a pathologic complete response (pCR) after NAC. Furthermore, we aimed to analyze the additional criteria for omitting PMRT in patients with a residual disease.

## 2. Materials and Methods

### 2.1. Study Populations

Breast cancer patients who underwent a total mastectomy after undergoing NAC were retrospectively identified from the Gangnam Severance Hospital and Severance Hospital breast cancer registries. All patients who were diagnosed with breast cancer between October 2006 and January 2018 were included. The database registry includes information on clinicopathologic characteristics, recurrence, and survival profiles. Axillary LN dissection was performed in all patients with sentinel LN metastasis, in accordance with the existing guidelines [22]. Patients who had undergone surgery before chemotherapy, or breast conservation surgery, as well as those with de novo stage IV disease, were excluded.

Next, the patients were selected using propensity score matching (PSM), as the pathologic characteristics of patients who underwent PMRT differed significantly from those of patients who did not undergo this treatment. Patients who did not receive PMRT were matched based on clinicopathologic variables, including age, clinical tumor/nodal (cT/N) stage, estrogen receptor (ER)/progesterone receptor (PR)/HER2 status, and pathologic stage (ypStage).

To compare the residual cancer burden (RCB) index and the neoadjuvant response index (NRI), another cohort from Gangnam Severance Hospital, with information on the RCB index, was used. These patients underwent surgery after NAC, between January 2007 and April 2020.

Our study was approved by the institutional review board of Gangnam Severance Hospital (approval number: 3-2021-0183), which waived the requirement for obtaining a written informed consent owing to the retrospective nature of the study.

### 2.2. Clinical and Pathologic Stage

We evaluated the patients’ initial cStage, prior to the administration of NAC, using mammography, ultrasonography, and breast magnetic resonance imaging. The clinical and pathologic stages were determined based on the tumor-node-metastasis classification [22]. However, the cN stage subclassification was modified, based on a definition similar to that of the ypN stage, using the number and location of metastatic LNs, as we were unable to determine whether the axillary LNs were movable or fixed. cN1 was defined as the presence of one to three metastatic LNs at axillary level I or II, while cN2 was defined as the presence of four or more metastatic LNs in the same position as cN1. Patients with metastases at axillary level III, or supraclavicular or internal mammary LNs, were classified as having cN3. If the results of each modality were not consistent, the higher stage was adopted as the cStage. pCR was defined as the absence of both invasive residual tumors in the breast, and axillary LN metastases [7].

### 2.3. Post-Mastectomy Radiotherapy

All patients for whom irradiation was planned underwent CT simulation before undergoing PMRT. The chest wall and nodal area were delineated according to the Radiation Therapy Oncology Group (RTOG) contouring guidelines [23]. Three-dimensional conformal PMRT was administered to the chest and regional nodal areas, including axillary levels I/II/III, and the supraclavicular internal mammary LNs, with a total radiation dose of 50.4 Gy, divided into 28 fractions.

### 2.4. NAC Response Index

The NRI is a semi-continuous scoring system that assesses the degree of downstaging induced by NAC [24]. The NRI was calculated using the cT/N stage, ypT/N stage, and breast pCR. The scores ranged from 0 to 1, with 1 indicating pCR and 0 indicating stable disease (SD) or disease progression (PD). A residual tumor size of <5 mm indicated a near breast pCR. A redefined cN stage was applied, and patients who did not undergo LN biopsy were also included. The method used to calculate the NRI is described in Appendix A.

The RCB index was another tool used for evaluating the response to NAC. It can be calculated using pathological variables, consisting of the primary tumor-bed size, the cellularity fraction of invasive cancer, the size of the largest metastatic LN, and the number of positive LNs [25]. The patients were divided into four groups, from RCB 0 to III. RCB 0 indicates pCR, while RCBs I, II, and III indicate minimal, moderate, and extensive tumor burden, respectively.

### 2.5. Statistical Analysis

The primary objective was to compare the difference in locoregional recurrence-free survival (LRRFS) between the groups that had, and had not, undergone PMRT. LRRFS was defined as the period between breast cancer diagnosis and the development of a new tumor in the skin, chest wall, or regional LNs, on the side previously affected by cancer. The secondary endpoint was the OS between the two groups. OS was defined as the period between breast cancer diagnosis and death from any cause. The differences between the groups were evaluated using the chi-square test. Continuous variables were compared using an independent two-sample *t*-test. Kaplan–Meier survival estimates were used to determine the impact of PMRT, and the group differences were analyzed using the log-rank test. All statistical tests were two sided, and a *p* value of <0.050 was considered significant. SPSS version 25.0 software (IBM Inc., Armonk, NY, USA) was used to perform all statistical analyses. The results of PSM were analyzed using SAS version 9.3 software (SAS Institute Inc., Cary, NC, USA).

## 3. Results

### 3.1. Baseline Characteristics

A total of 914 patients were followed, for a median of 56 months, after diagnosis. Of these patients, 777 (85.0%) received PMRT, while 137 (15%) did not receive this treatment. The 5-year LRRFS and 5-year OS rates were 97.2% and 92.1%, respectively, in all patients. Table 1 summarizes the clinicopathological characteristics of all patients. The average age of the patients was 48.2 years. The mean initial estimated tumor size was 44 mm. A total of 849 (92.9%) patients were classified as having cN+. A total of 187 (20.5%) patients achieved pCR, while 204 (22.3%) patients still had ypStage III tumors after undergoing NAC. Approximately 82.7% of the patients received anthracycline and taxane-based chemotherapy, while 41.1% of the HER2-positive patients received anti-HER2 therapy. Patients who underwent PMRT were more likely to be classified as having cN+ than those who did not undergo PMRT. ER-positive breast cancer was more common in the PMRT group; patients with HER2-negative tumors more frequently underwent PMRT than those with HER2-positive tumors. The pCR rate was significantly higher in patients who did not undergo PMRT. The pCR rate was the highest in patients with HER2-positive tumors (34.6%), but was significantly lower in patients with ER-positive/HER2-negative tumors (6.5%; Appendix A).

After 1:3 PSM, 368 patients were selected. Of these patients, 276 (75%) and 92 (25%) comprised the PMRT and non-PMRT groups, respectively. No significant differences were found in the pathologic features between the two groups after PSM (Table 2).

### 3.2. Impact of PMRT in pCR and Non-pCR Patients

In patients who underwent PMRT, the 5-year LRRFS rate was 97.3%, while the 5-year OS rate was 92.4%. In patients who did not undergo PMRT, the 5-year LRRFS rate was 96.4%, while the 5-year OS rate was 90.5%. No significant difference was observed in the prognosis between the two groups (LRRFS, *p* = 0.185; OS, *p* = 0.128, Appendix A). When the patients were divided according to pCR status, the pCR group showed better LRRFS and OS rates than the non-pCR group (LRRFS; *p* = 0.031; OS: *p* = 0.013, Appendix A). In the multivariable analysis, adjusted for age, cT/N stage, ER/PR/HER2 status, and ypStage values, the LRRFS and OS rates were significantly better among patients who underwent PMRT (LRRFS, hazard ratio (HR): 0.27, 95% confidence interval (CI): 0.11–0.71, *p* = 0.008; OS, HR: 0.38, 95% CI: 0.21–0.68, *p* = 0.001; Appendix A. However, in the pCR group, PMRT was not associated with LRR or survival outcome (LRRFS, not estimated (NE); OS, HR: 0.47, 95% CI: 0.08–2.79, *p* = 0.409). PMRT only benefitted those in the non-pCR group (LRRFS, HR: 0.24, 95% CI: 0.09–0.62, *p* = 0.003; OS, HR: 0.34, 95% CI: 0.18–0.63, *p* = 0.001).

In the matched patients, PMRT had a significant positive influence on the LRRFS (*p* = 0.016; Figure 1A). OS was also improved in patients who underwent PMRT compared to those who did not undergo this treatment (*p* = 0.017; Figure 1B). Next, the patients selected by PSM were divided into pCR and non-pCR groups. The pCR group was comprised of 77 patients (20.9%), while the non-pCR group was comprised of 291 patients (79.1%). Among those who achieved pCR, no differences were observed in the LRRFS and OS rates between patients who received PMRT and those who did not receive PMRT (LRRFS, *p* = 0.999; OS, *p* = 0.453; Figure 2A,B). By contrast, PMRT significantly improved the LRRFS and OS rates in the non-pCR group (LRRFS, *p* = 0.012; OS, *p* = 0.006; Figure 2C,D).

### 3.3. Effect of PMRT According to Patient’s Response to NAC Using Neoadjuvant Response Index

To confirm the accuracy of NRI, the results obtained using the RCB index and NRI were compared with the results of another cohort of patients, from the Gangnam Severance Hospital, whose RCB index values were obtained. Patients with RCB 0 or I were considered good responders, while those with RCB II or III were classified as poor responders [23]. Patients with 0.7 < NRI ≤ 1 were defined as good responders [24,26]. Among 383 patients with missing data on RCB index and NRI values, 353 (92.2%) had the same RCB index and NRI values (Appendix A). A total of 198 patients had RCB 0 or I, while 179 (90.4%) of them were good responders according to their NRI values. Among the 185 patients who had RCB II or III, 174 (94.1%) were poor responders according to their NRI values. Only 30 patients had different outcomes: 11 (2.9%) had a higher RCB index, and 19 (5.0%) had a higher NRI.

The distribution of all patients and PSM patients according to their NRI values is summarized in Appendix A. Among the good responders, 234 (86.3%) had ypN0 tumors. Although metastatic LNs remained in 37 (13.7%) patients, they achieved breast pCR. In this group, LRRFS and OS rates did not differ between patients who did and did not undergo PMRT, according to the results of the multivariable analysis (LRRFS and OS: NE, Appendix A). PMRT only benefited patients with poor treatment response (LRRFS, HR: 0.21, 95% CI: 0.08–0.57, *p* = 0.002; OS, HR: 0.33, 95% CI: 0.17–0.62, *p* = 0.001). In 1:3 PSM patients, PMRT was not a significant prognostic factor in the good responder group (LRRFS, *p* = 0.568; OS, *p* = 0.875; Figure 3A,B). Furthermore, PMRT did not benefit patients with 0.7 < NRI < 1, excluding the patients with pCR (Appendix A). However, in the poor responder group, patients who received PMRT had significantly better LRRFS and OS rates than those who did not receive this treatment (LRRFS, *p* = 0.011; OS, *p* = 0.006; Figure 3C,D).

### 3.4. Subgroup Analysis of Survival Outcomes According to PMRT Status

A subgroup analysis was performed to determine the effect of PMRT on OS in 1:3 PSM patients (Figure 4). When the results were validated according to ypStage in the non-pCR group, PMRT had a more significant effect on OS in patients with ypStage II or III than in those with ypStage I (HR: 0.30, 95% CI: 0.14–0.63, *p* = 0.001). The effect of PMRT was not significant in patients whose LN metastases were resolved after receiving NAC (HR: 1.07, 95% CI: 0.21–5.50, *p* = 0.939). Conversely, in patients whose LN metastases remained after undergoing chemotherapy, PMRT improved their survival rates (HR: 0.25, 95% CI: 0.12–0.54, *p* < 0.001). Among patients who were diagnosed with locally advanced breast cancer, PMRT did not affect patients with pCR/ypStage I (HR: 1.40, 95% CI: 0.16–12.61, *p* = 0.763). However, PMRT had a significant beneficial effect on OS (HR: 0.25, 95% CI: 0.11–0.56, *p* = 0.001) in patients with ypStage II/III. Other subgroups, according to cT stage or breast tumor subtypes, did not exhibit significant differences in PMRT effects. Similar trends were observed in the results of the subgroup analysis of LRRFS (Appendix A).

## 4. Discussion

Our study showed that radiotherapy was a useful treatment for patients who underwent total mastectomy after NAC. PMRT should be recommended in patients who have large residual disease after chemotherapy. Conversely, the benefit of PMRT was not clear in patients who achieved pCR. In addition, we found a group of good responders who did not need PMRT, despite not achieving pCR. This result confirmed that the patients’ response to NAC is important when determining which patients require, or do not require, PMRT.

After performing PSM to adjust for various factors that could influence prognosis, the benefits of PMRT were evaluated in patients with NAC. Krug et al. stated that PMRT only reduced the LRR rate, and did not improve the disease-free survival rate, after investigating a cohort in three neoadjuvant prospective trials (GeparTrio, GeparQuattro, and GeparQuinto) [19]. However, our results showed that PMRT had a positive influence on both LRR and OS. This discrepancy is probably due to the significantly higher proportion of cN0 patients in the previous study (37.8%) compared with our study (7.1%).

pCR is a strong prognostic factor in NAC patients [6,7,27]. Our results are consistent with the finding of these studies, showing that pCR can improve patient’s prognosis. We hypothesized that the effect of PMRT could differ depending on whether or not patients achieved pCR. Previous studies investigated the effect of PMRT in patients with pCR. Krug et al. demonstrated no significant difference in LRR following PMRT in patients who achieved pCR [19]. On the contrary, McGuire et al. argued that PMRT provides clinical benefits to patients with cStage III disease who achieved pCR [28]. However, this was due to the small number of patients and the lack of adjustment for clinicopathologic factors in their study. We reported that PMRT had a clinical benefit on LRR and survival, in non-pCR patients only, in a comparatively large population, by correcting various factors. Moreover, PMRT was not helpful in patients with ypStage I, or who had achieved pCR, even if they had cStage III prior to the administration of NAC.

However, with the binary outcome of pCR, important information regarding patient’s response to NAC might be missed. Therefore, many NAC response indices, such as RCB index and NRI, have recently been developed. In this study, NRI was used to evaluate NAC response, despite RCB being more widely used, because pathologic evaluation is essential to obtain the RCB index; however, this information was not available for most of our patients. Instead, the RCB index and NRI were compared in a patient group with available RCB index value. As the results obtained using the NRI were consistent with those obtained using the RCB index, we determined that the NRI could replace the RCB index in this study. In the future, the benefits of PMRT should be validated using the RCB index.

To the best of our knowledge, this was the first study to analyze the impact of PMRT according to the NAC response index. A previous study demonstrated that the survival outcome of patients with minimal residual disease, according to the RCB index and NRI, was non-inferior to that of patients with pCR [26]. In patients with minimal residual disease, one must determine whether PMRT can help improve the patient’s prognosis. No significant difference was found in LRRFS and OS rates, based on the status of PMRT use, in patients with a good response to NAC. Thus, our study suggests that the NAC response index, including NRI, can be a criterion for deciding the necessity of administering PMRT in non-pCR patients.

Several previous studies have determined a subpopulation of patients who benefited from PMRT, among those who received NAC. Scodan et al. and Cho et al. suggested that PMRT is beneficial in patients with ypN+ disease after receiving NAC, but not in those with ypN0 disease [21,29]. Furthermore, some researchers argued that PMRT can also be omitted in patients with ypN1 disease [30,31]. Conversely, Rusthoven et al. and Krug et al. reported that PMRT is helpful in ypN0 patients if the patient was initially classified as having a cN+ disease [18,19]. However, as the cohort of Rusthoven et al. [18], from the National Cancer Database, included patients who received radiotherapy only in the chest wall, without regional LN irradiation, the definition of the PMRT field was unclear. In addition, the results of Krug et al. [19] did not show a clear difference, and had only a marginal statistical value (*p* = 0.050). Our study confirmed that, even if the patients had LN metastases prior to the administration of chemotherapy, PMRT did not benefit patients who were classified as having ypN0 after receiving NAC, among those who received consistent radiotherapy. Among patients with locally advanced breast cancer, PMRT only significantly affected those with large residual disease. Therefore, the results of current prospective trials, such as the NRG oncology/National Surgical Adjuvant Breast and Bowel Project (NSABP) B-51/RTOG 1304, which compare the outcomes of patients with and without PMRT after receiving NAC, should be awaited.

This study has some limitations. It is retrospective in nature, and our data may have a selection bias. To compensate for this, two homogeneous study groups were made through PSM. In addition, our follow-up period was relatively short. Further studies are required to analyze the information of patients with a longer follow-up period. As mentioned above, information on the RCB index cannot be used due to various restrictions. However, the NRI is also one of the NAC response indices that have recently been recognized as excellent. Furthermore, by comparing the NRI and RCB index of our cohort with those of another cohort, we confirmed that the results were consistent.

## 5. Conclusions

Radiotherapy is useful for treating patients who underwent total mastectomy after NAC. However, the benefit of PMRT may vary depending on the patients’ response to NAC. Clinicians can omit PMRT not only in patients with pCR, but also in good responders with an NRI value of 0.7–1.0. The results from the NRG oncology/NSABP B-51/RTOG 1304 trial (ClinicalTrials.gov Identifier: NCT01872975), an ongoing prospective study, will further clarify the criteria for determining the need to perform PMRT after NAC.

## Figures and Tables

**Figure 1 cancers-13-06205-f001:**
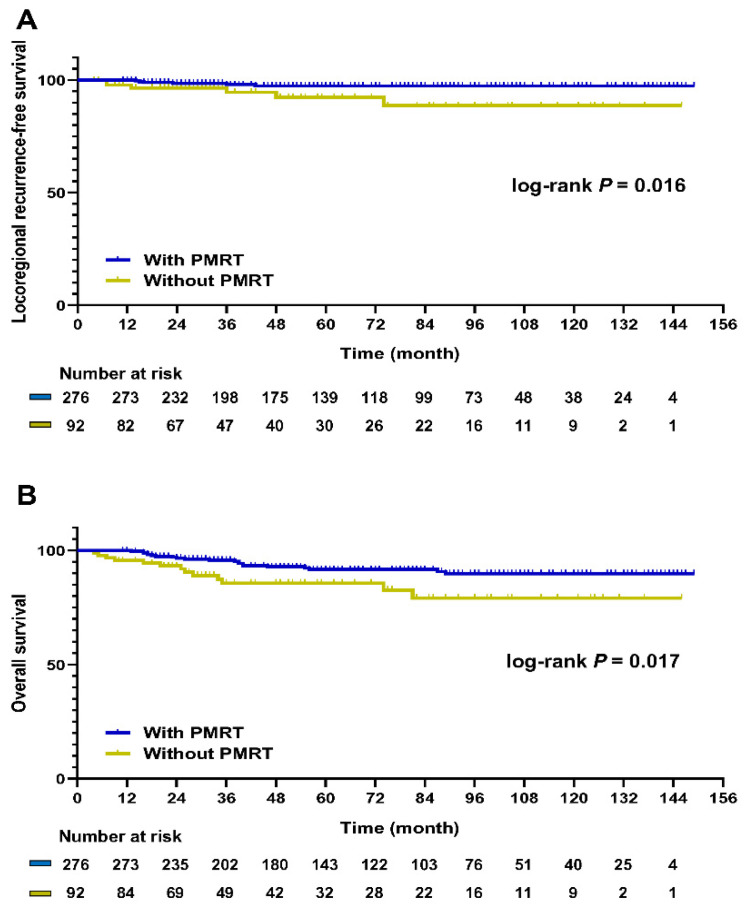
Kaplan–Meier survival curve of LRRFS and OS according to PMRT status after 1:3 PSM. (**A**) LRRFS (*p* = 0.016); (**B**) OS (*p* = 0.017). LRRFS, locoregional recurrence-free survival; OS, overall survival; PMRT, post-mastectomy radiotherapy; PSM, propensity score matching.

**Figure 2 cancers-13-06205-f002:**
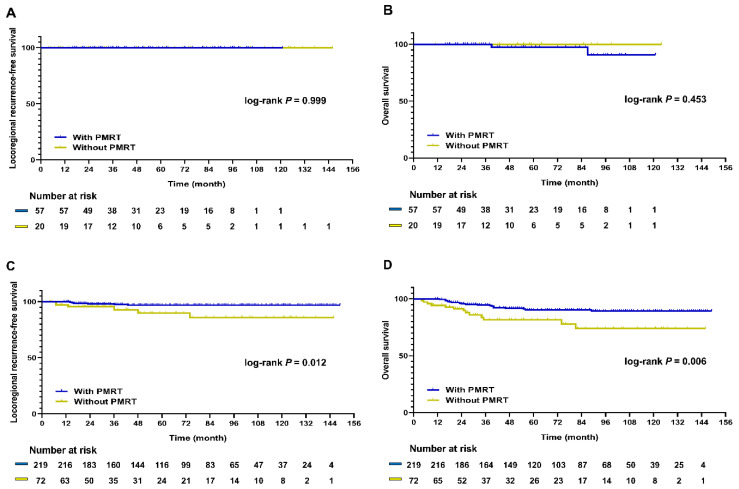
Kaplan–Meier survival curve of LRRFS and OS according to PMRT and pCR status after 1:3 PSM. (**A**) LRRFS in the pCR group (*p* = 0.999); (**B**) OS in the pCR group (*p* = 0.453); (**C**) LRRFS in the non-pCR group (*p* = 0.012); (**D**) OS in the non-pCR group (*p* = 0.006). LRRFS, locoregional recurrence-free survival; OS, overall survival; PMRT, post-mastectomy radiotherapy; pCR, pathologic complete response; PSM, propensity score matching.

**Figure 3 cancers-13-06205-f003:**
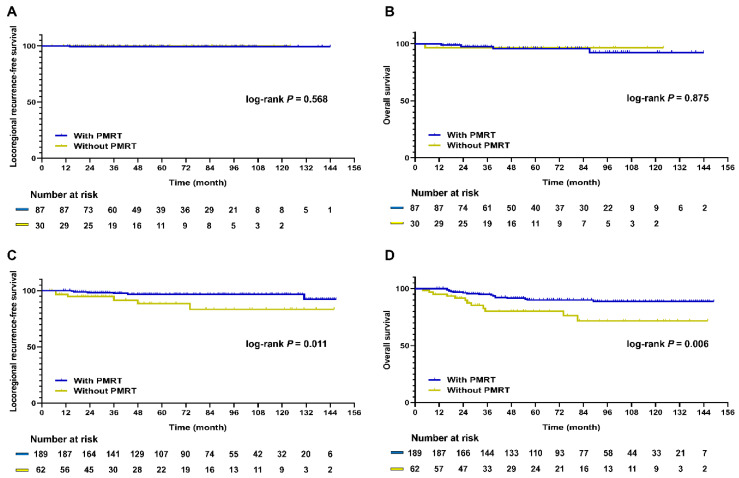
Kaplan–Meier survival curve of LRRFS and OS according to PMRT status in 1:3 PSM patients. (**A**) LRRFS in patients with 0.7 < NRI ≤ 1 (*p* = 0.568); (**B**) OS in patients with 0.7 < NRI ≤ 1 (*p* = 0.875); (**C**) LRRFS in patients with NRI < 0.7 (*p* = 0.011); (**D**) OS in patients with NRI < 0.7 (*p* = 0.006). LRRFS, locoregional recurrence-free survival; OS, overall survival; PMRT, post-mastectomy radiotherapy; NRI, neoadjuvant response index; PSM, propensity score matching.

**Figure 4 cancers-13-06205-f004:**
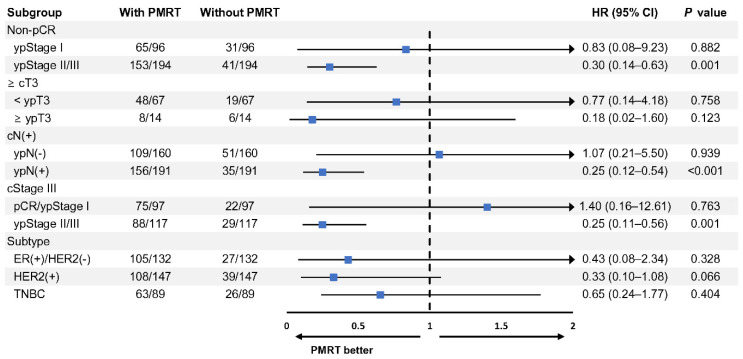
Subgroup analysis of OS according to PMRT status in 1:3 PSM patients. OS, overall survival; PSM, propensity score matching; PMRT, post-mastectomy radiotherapy; HR, hazard ratio; CI, confidence interval; pCR, pathologic complete response; ER, estrogen receptor; HER2, human epidermal growth factor receptor 2; TNBC, triple-negative breast cancer.

**Table 1 cancers-13-06205-t001:** Baseline characteristics of all patients with or without PMRT.

Variable	All Patients (%)	Patients with PMRT (%)	Patients without PMRT (%)	*p* Value
Total	914 (100)	777 (100)	137 (100)	-
Age at diagnosis, average (range), years	48.2 (20–79)	48.0 (20–79)	49.3 (27–76)	0.295
cT stage	-	-	-	0.057
cT1	115 (12.6)	103 (13.3)	12 (8.8)	-
cT2	544 (59.5)	450 (57.9)	94 (68.6)	-
cT3–4	255 (27.9)	224 (28.8)	31 (22.6)	-
cN Stage	-	-	-	<0.001
cN0	65 (7.1)	38 (4.9)	27 (19.7)	-
cN1	414 (45.3)	362 (46.6)	52 (38.0)	-
cN2	286 (31.3)	242 (31.1)	44 (32.1)	-
cN3	149 (16.3)	135 (17.4)	14 (10.2)	-
Estrogen receptor	-	-	-	0.001
Positive	530 (58.0)	469 (60.4)	61 (44.5)	-
Negative	384 (42.0)	308 (39.6)	76 (55.5)	-
Progesterone receptor	-	-	-	0.009
Positive	388 (42.5)	344 (44.3)	44 (32.1)	-
Negative	526 (57.5)	433 (55.7)	93 (67.9)	-
HER2	-	-	-	0.002
Negative	576 (63.0)	506 (65.1)	70 (51.1)	-
Positive	338 (37.0)	271 (34.9)	67 (48.9)	-
Chemotherapy regimen	-	-	-	0.562
Anthracycline based	57 (6.2)	49 (6.3)	8 (5.8)	-
Taxane based	96 (10.5)	78 (10.0)	18 (13.1)	-
Anthracycline and taxane based	756 (82.7)	645 (83.0)	111 (81.0)	-
Unknown	5 (0.5)	5 (0.6)	0 (0)	-
Anti-HER2 therapy in HER2-positive	-	-	-	0.339
Performed	139 (41.1)	108 (39.9)	31 (46.3)	-
Not performed	199 (58.9)	163 (60.1)	36 (53.7)	-
ypStage	-	-	-	<0.001
pCR	187 (20.5)	135 (17.4)	50 (36.5)	-
Stage I	219 (24.0)	182 (23.4)	39 (28.5)	-
Stage II	304 (33.3)	270 (34.7)	34 (24.8)	-
Stage III	204 (22.3)	190 (24.5)	14 (10.2)	-

PMRT, post-mastectomy radiotherapy; HER2, human epidermal growth factor receptor 2; pCR, pathologic complete response.

**Table 2 cancers-13-06205-t002:** Clinicopathologic characteristics of patients after 1:3 PSM.

Variable	Patients with PMRT (%)	Patients without PMRT (%)	*p* Value
Total	276 (100)	92 (100)	-
Age at diagnosis, median (range), years	47.8 (26–79)	49.0 (27–74)	0.335
cT stage	-	-	0.386
cT1	33 (12.0)	10 (10.9)	-
cT2	187 (67.8)	57 (62.0)	-
cT3–4	56 (20.3)	25 (27.2)	-
cN stage	-	-	0.210
cN0	11 (4.0%)	6 (6.5%)	-
cN1	111 (40.2%)	39 (42.4%)	-
cN2	103 (37.3%)	38 (41.3%)	-
cN3	51 (18.5%)	9 (9.8%)	-
Estrogen receptor	-	-	0.547
Positive	149 (54.0)	46 (50.0)	-
Negative	127 (46.0)	46 (50.0)	-
Progesterone receptor	-	-	0.619
Positive	106 (38.4)	32 (34.8)	-
Negative	170 (61.6)	60 (65.2)	-
HER2	-	-	0.624
Negative	168 (60.9)	53 (57.6)	-
Positive	108 (39.1)	39 (42.4)	-
Chemotherapy regimen	-	-	0.935
Anthracycline based	16 (5.8)	5 (5.4)	-
Taxane based	32 (11.7)	12 (13.0)	-
Anthracycline and taxane based	226 (82.5)	75 (81.5)	-
Anti-HER2 therapy in HER2 positive	-	-	0.367
Performed	42 (36.7)	12 (30.8)	-
Not performed	66 (61.1)	27 (69.2)	-
ypStage	-	-	0.191
pCR	57 (20.7)	20 (21.7)	-
Stage I	65 (23.6)	31 (33.7)	-
Stage II	97 (35.1)	28 (30.4)	-
Stage III	57 (20.7)	13 (14.1)	-

PSM, propensity score matching; PMRT, post-mastectomy radiotherapy; HER2, human epidermal growth factor receptor; pCR, pathologic complete response.

## Data Availability

The datasets used and/or analyzed during the current study are available from the corresponding author, on reasonable request.

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
