# Peer review of "The Impact of Post-Mastectomy Radiotherapy on Survival Outcomes in Breast Cancer Patients Who Underwent Neoadjuvant Chemotherapy"

_cancers, 2021, doi:10.3390/cancers13246205_

Round 1

Reviewer 1 Report

Very good subject to support lower therapy ( no radiotherapy) in selected case of pCR after NAC when mastectectomy and axillary lymphnode dissection is done 

Very good article :  the methodology is correct even if it is a retrospective study and the authors clearly specify in discussion part the notes inherent in this level of evidence wich is remains low . The material et methods part must be placed before the results . Results well presented. Discussion interesting and well written. Good conclusion wich opens and explains the future prospects for these patients  

Author Response

Very good subject to support lower therapy ( no radiotherapy) in selected case of pCR after NAC when mastectectomy and axillary lymphnode dissection is done 

Very good article :  the methodology is correct even if it is a retrospective study and the authors clearly specify in discussion part the notes inherent in this level of evidence wich is remains low . The material et methods part must be placed before the results . Results well presented. Discussion interesting and well written. Good conclusion wich opens and explains the future prospects for these patients 

R: Thank you for the good review of our manuscript. Regarding the part you commented “Extensive editing of English language and style required”, additional English correction was carried out.

Reviewer 2 Report

this a retrospective analysis  on the effect of PMRT in patients after NACT. 368 patients were included in a 1:3 propensity score matching. The topic is highly relevant since most guidelines recommend PMRT based on data from a metaanalysis from EBCTCG including patients treated between the 60ies and 80ies of last century. In view of the increasing use of NACT it appears important to identify more individualized factors for risk assesment, especially as the primary lymph node status is not available for many cN0 patients, who undergo SLNB after NACT. Response to NACT has been proven to be a strong prognostic factor. The question whether response could be used as a tool to tailor PMRT is highly relevant.

The authors showed in their series (92 pts without PMRT, 276 with PMRT) that there was a significant relation between response and locoregional recurrence free survival and overall survival). Patients with pCR or good but incomplete response had no benefit from PMRT in contrast to patients with less response assessed by the neoadjuvant response index.

These results underline the hypothesis from Mamounas et al from data of a joint analysis of B18 and B27. The concept is currently investigated in a prospective randomized study (B51). The literature is correctly discussed.

I have a few comments for improvement:

  1. The section Material and Methods should be the second chapter, prior to the results
  2. It should be made clear, that all patients recieved comparable chemotherapy regimen (related to drugs and cycles)
  3. Please be more careful in rating your results: The study did not "reveal" (line 173) or "prove" (line 194) but rather support the hypothesis that ...
  4. weaknesses of the study are the retrospective design, relatively low patient numbers and possible selection bias (in spite of PSM). These weaknesses have, however been adaequately discussed including all relevant literature
  5. A strenght of the paper is that this the first study that examines the PMRT effect not only with with to pCR vs not not but also to in relation to different response grades

Author Response

this a retrospective analysis  on the effect of PMRT in patients after NACT. 368 patients were included in a 1:3 propensity score matching. The topic is highly relevant since most guidelines recommend PMRT based on data from a metaanalysis from EBCTCG including patients treated between the 60ies and 80ies of last century. In view of the increasing use of NACT it appears important to identify more individualized factors for risk assesment, especially as the primary lymph node status is not available for many cN0 patients, who undergo SLNB after NACT. Response to NACT has been proven to be a strong prognostic factor. The question whether response could be used as a tool to tailor PMRT is highly relevant.

The authors showed in their series (92 pts without PMRT, 276 with PMRT) that there was a significant relation between response and locoregional recurrence free survival and overall survival). Patients with pCR or good but incomplete response had no benefit from PMRT in contrast to patients with less response assessed by the neoadjuvant response index.

These results underline the hypothesis from Mamounas et al from data of a joint analysis of B18 and B27. The concept is currently investigated in a prospective randomized study (B51). The literature is correctly discussed.

I have a few comments for improvement:

  1. The section Material and Methods should be the second chapter, prior to the results

R: Thank you for your precise comment. We have rearranged the order of the sections so that “Material and Methods” is in the second chapter.

  1. It should be made clear, that all patients recieved comparable chemotherapy regimen (related to drugs and cycles)

R: Thank you for your precise comment. We attached information on chemotherapy regimens to the clinicopathologic characteristics table for all patient and 1:3 PSM patients. Regimens were largely divided into anthracycline-based, taxane based and anthracycline and taxane-based. In addition, it was found that most patients in our cohort had completed the initially planned cycle of chemotherapy. However, there was a limitation in showing this in summary because the number of cycles determined by each regimen is different.

  1. Please be more careful in rating your results: The study did not "reveal" (line 173) or "prove" (line 194) but rather support the hypothesis that ...

R: Thank you for your comment. We have changed the expression of the word you pointed out.

  1. weaknesses of the study are the retrospective design, relatively low patient numbers and possible selection bias (in spite of PSM). These weaknesses have, however been adaequately discussed including all relevant literature.

R: Thank you for the good review of our manuscript.

  1. A strenght of the paper is that this the first study that examines the PMRT effect not only with with to pCR vs not not but also to in relation to different response grades

R: Thank you for your precise comment. As you note, we have slightly modified our conclusions to further highlight the strengths of our study.

Reviewer 3 Report

The authors performed a retrospective analysis of 368 patients treated with postmastectomyradiotherapy (PMRT) after neoadjuvant chemotherapy (NACT) between 2006 and 2018 in a single institution, in order to determine whether PMRT is beneficial in patients with pathologic complete response (pCR) or with minimal residual disease. Patients who did not receive PMRT were matched 1:3 (propensity score matching) and locoregional recurrence-free survival and overall survival were compared. PMRT improved survival only in patients with large residual tumor burden, such as nodal metastases and pathologic stage II/III, and not in patients with pCR or minimal residual disease. The authors recognized and mentioned the limitations of the study: its retrospective nature, the possible selection bias ad the short follow-up. In the absence of solid prospective evidence of the efficacy of the PMRT after NACT, there remains a role for retrospective studies. In the opinion of this reviewer, the study does not have a very large number of patients, and the follow-up is less than a minimum of 5 years. Its results are in line with those publised by other similar studies.

Material and methods should be the second chapter, after Introduction and followed by Results, Discussion and Conclusion. Please make the change. 

I would like to thank the editor for allowing me to review the paper.

Author Response

The authors performed a retrospective analysis of 368 patients treated with postmastectomyradiotherapy (PMRT) after neoadjuvant chemotherapy (NACT) between 2006 and 2018 in a single institution, in order to determine whether PMRT is beneficial in patients with pathologic complete response (pCR) or with minimal residual disease. Patients who did not receive PMRT were matched 1:3 (propensity score matching) and locoregional recurrence-free survival and overall survival were compared. PMRT improved survival only in patients with large residual tumor burden, such as nodal metastases and pathologic stage II/III, and not in patients with pCR or minimal residual disease. The authors recognized and mentioned the limitations of the study: its retrospective nature, the possible selection bias ad the short follow-up. In the absence of solid prospective evidence of the efficacy of the PMRT after NACT, there remains a role for retrospective studies. In the opinion of this reviewer, the study does not have a very large number of patients, and the follow-up is less than a minimum of 5 years. Its results are in line with those publised by other similar studies.

Material and methods should be the second chapter, after Introduction and followed by Results, Discussion and Conclusion. Please make the change. 

R: Thank you for your precise comment. We have rearranged the order of the sections so that “Material and Methods” is in the second chapter.

Round 2

Reviewer 2 Report

thank you for revising the manuscript. For me this is an important contribution that certainly needs confirmation by a randomized trial